# Cell-Free Protein Synthesis by Diversifying Bacterial Transcription Machinery

**DOI:** 10.3390/biotech10040024

**Published:** 2021-10-14

**Authors:** Marina Snapyan, Sylvain Robin, Garabet Yeretssian, Michèle Lecocq, Frédéric Marc, Vehary Sakanyan

**Affiliations:** 1UMR CNRS 6204, Faculté des Sciences et des Techniques, Université de Nantes, 44322 Nantes, France; marina.snapyan1@ulaval.ca (M.S.); gyeretssian78@gmail.com (G.Y.); michelle.lecocq@univ-nantes.fr (M.L.); 2ProtNeteomix, 44322 Nantes, France; sylva_rob@yahoo.fr (S.R.); frederic.marc-kreucher@merckgroup.com (F.M.); 3IICiMed, Faculté de Pharmacie, Université de Nantes, 44322 Nantes, France

**Keywords:** cell-free protein synthesis, *rpo* genes coexpression, RNA polymerase, bacterial promoter, RNase I, protein overexpression

## Abstract

We have evaluated several approaches to increase protein synthesis in a cell-free coupled bacterial transcription and translation system. A strong p*argC* promoter, originally isolated from a moderate thermophilic bacterium *Geobacillus stearothermophilus*, was used to improve the performance of a cell-free system in extracts of *Escherichia coli* BL21 (DE3). A stimulating effect on protein synthesis was detected with extracts prepared from recombinant cells, in which the *E. coli* RNA polymerase subunits α, β, β’ and ω are simultaneously coexpressed. Appending a 3′ UTR genomic sequence and a T7 transcription terminator to the protein-coding region also improves the synthetic activity of some genes from linear DNA. The *E. coli* BL21 (DE3) *rna*::Tn10 mutant deficient in a periplasmic RNase I was constructed. The mutant cell-free extract increases by up to four-fold the expression of bacterial and human genes mediated from both bacterial p*argC* and phage pT7 promoters. By contrast, the RNase E deficiency does not affect the cell-free expression of the same genes. The regulatory proteins of the extremophilic bacterium *Thermotoga*, synthesized in a cell-free system, can provide the binding capacity to target DNA regions. The advantageous characteristics of cell-free systems described open attractive opportunities for high-throughput screening assays.

## 1. Introduction

A cell-free method for protein production has been successfully implemented by translating exogenously added nucleic acid into cell extracts of *Escherichia coli* [1]. Protein synthesis using the T7 phage transcriptional machinery increased protein yield and turned cell-free protein synthesis (CFPS) into a powerful approach for studying the proteomes of organisms with sequenced genomes without performing laborious gene cloning in living cells. The single subunit T7 RNA polymerase catalyzes mRNA synthesis with an elongation rate of 200 nucleotides per second, 4–8-fold higher than the multisubunit RNA polymerase of *E. coli* [2]. However, bacterial transcriptional machinery is conserved in distant bacteria [3], and its RNA polymerase can recognize various heterologous promoters, constituting a great advantage in the use of native transcriptional signals for in vitro gene expression studies. Similar CFPS systems have also been created using other bacterial species to achieve additional advantages over the *E. coli* system for detecting specific protein–protein interactions and identifying potential inhibitors of desired proteins [4,5].

Nine endonucleases have been identified and studied in *E. coli*, and the individual activity of these enzymes is responsible for mRNA degradation [6]. The protein yield depends not only on the rate of transcription but also on the stability of the target mRNAs synthesized, a question that is still obscure regarding cell-free systems. It is generally accepted that the fate of nascent mRNAs in bacteria depends on the activity of a multiprotein complex, the degradosome, in which RNase E plays a major role. Frameshift mutations, leading to the truncation of a C-terminal region of RNase E, decrease the RNase activity without affecting RNA processing and host cell growth [7]. Another enzyme, RNase I, is responsible for cleavage at sites located within full-length mRNAs [8,9]. The thermostable RNase I is in the periplasm of Gram-negative bacteria and accounts for more than 98% of total RNase activity in crude extracts assayed in the presence of EDTA [10]. The enzyme attacks also double-strand RNA structures formed in any type of cellular RNA at the increased ratio of enzyme/RNA. In view of the RNase I, which is liberated during the preparation of cell-free extracts, the question of the impact on mRNA degradation becomes important to improve the performance of CFPS.

We have shown that a p*argC* strong promoter (Figure 1A), preceding the *Geobacillus stearothermophilus argCJBD* operon, can be applied to CFPS with extracts of *E. coli* [11]. A logical extension of this finding is to improve the p*argC*-mediated gene expression with the aim of exploiting it for future proteomic research. In this study, several approaches were tested to enhance protein productivity in a cell-free system. We improved the yield of both prokaryotic and eukaryotic proteins by the integration of strong bacterial and phage transcription signals into the cell-free extracts derived from mutant *E. coli* deficient in the RNase I activity.

## 2. Materials and Methods

### 2.1. DNA Cloning

The *E. coli* O157:H7 EDL933 *rpo* genes were cloned into two compatible vectors, pETDuet-1 and pACYCDuet-1 [12], each designed for the cloning and expression of two target genes (Merck, Lyon, France). The *rpoA*/*rpoB* and *rpoC*/*rpoZ* coding for couples of α/β and β’/ω subunits were amplified by PCR using Taq DNA polymerase (Qiagen, CourtaBoeuf, Villebon-sur-Yvette, France) and inserted, respectively, into pETDuet-1 and pACYCDuet-1 vectors to provide coexpression of RNA polymerase subunits in the *E. coli* BL21 (DE3) strain (Figure 1).

The strain *E. coli* B (DE3) was used for gene expression assays. Chromosomal DNA of hyperthermophile bacteria *Thermotoga maritima* MSB8 and *Thermotoga neapolitana* DSM5068 was isolated as described previously [13]. Assuming a similarity to *T. maritima* [14], the *T. neapolitana groESL* chaperonin region was amplified using degenerate oligonucleotide primers 5′-GGAGGGATGGATGATGAARGTNA (forward) and 5′-GAYTTYATHGAYCARCAYCCNGA (reverse) and inserted into pET19b between NdeI and BamHI restriction sites. The sequence *groES* was assigned EMBL accession AF275319. Other primers used for PCR amplification of *T. maritima* genes are described in Appendix A.

The *rraA* gene coding for the RNase E inhibitor protein RraA [15] was amplified by PCR from *E. coli* O157: H7 EDL 933 and inserted into expression vector pET28b+ between BamHI and HindIII restriction sites.

Comparison of cDNA and genomic DNA clones showed that the open reading frames of each of the NAT1 and NAT2 genes do not have introns, and the protein coding sequences are more than 80% identical [16]. Therefore, these genes were amplified by PCR on chromosomal DNA using forward primer 5′GATCATGGACATTGAAGCATATTTTGAAAG for NAT1 and NAT2 (initiation codon underlined) and reverse primers 5′CCTTATTCTAAATAGTAAAAAATCTATCAC for NAT1 and 5′CCTTATTCTAAATAGTAAGGGATCCATCACC for NAT2 (stop codon underlined). The genes were inserted into pET28b+ between NdeI and NotI restriction sites.

DNA concentration was determined by spectrophotometry and confirmed by comparing band intensity with a Smart Ladder reference DNA (Eurogentec, Liege, Belgium) after electrophoresis. The nucleotide sequences of PCR amplified genes were verified (Eurofins, Les Ulis, France).

### 2.2. Construction of Linear DNA Templates

The bacteriophage pT7 promoter and p*argC* promoter of *G. stearothermophilus argC* gene coding *N*-acetylglutamate-5-semialdehyde dehydrogenase [17] were used for driving the cell-free synthesis of target genes. A forward primer (5′-CATAGACTTAGGGAGGGGC) and a reverse primer (5′-ATGATGATGATGATGATGCATATGTTCCCCCTCACCCGT) were used to amplify the integral region covering the p*argC* promoter and the *argC* initiation codon (see Figure 1A). The reverse primer was designed to introduce 6xHis codons, creating the N-terminal tag necessary for further purification of the protein RpoA (α subunit of RNA polymerase) by affinity chromatography with Ni-NTA resin. The promoter p*argC* region was linked with the *G. stearothermophilus* gene *amaA* coding a 43 kDa *N*-acyl-amino acid amidohydrolase [18] or the *T. maritima gntR 0439* gene coding for a 25 kDa putative GntR regulator [19]. The connection of two DNA fragments was performed by the second-round PCR as described previously [20]. The quantity of DNA was quantified by lab-on-chip DNA 7500 assay kit with 2100 Bioanalyzer (Agilent Technologies, Les Ulis, France) by injecting a PCR product.

### 2.3. P1 Transduction

A bacteriophage P1 cml clr-100-mediated transduction of bacterial genes was performed at the multiplicity of infection equal to 0.1 [21]. The *rna*::Tn10 (kanamycin resistance) mutant gene [22] was transduced from the *E. coli* K12 DK533 strain to *E*. *coli* BL21 (DE3) and *E. coli* BL21 Star (DE3) (*rna*E131, deficient for RNase E). The transfer of a ~2 kb *rna*::Tn10 DNA region was confirmed by PCR amplification using oligonucleotide primers 5′-ATGAAAGCATTCTGGCGTAAC and 5′-TTAATAACCCGCTTTATCAATC corresponding to the beginning and the end of the RNase I.

### 2.4. Ribonuclease Assay

The reaction was carried out at 37 °C for 30 min with RNaseAlert Lab Test kit (Ambion, Berlin, Germany). Real-time fluorescence measurements were performed during incubation at excitation 490 nm and emission 520 nm with a fluorometer FP6500 (Jasco, Lisses, France) according to the supplier’s recommendations. A relative RNase I activity was expressed in % after 30 min incubation using the maximal activity detected in one of the extracts analyzed as a reference.

### 2.5. Purification of Proteins and Western Blotting

*E. coli* strains, in which T7 bacteriophage RNA polymerase synthesis is regulated by a *lacUV5* inducible promoter, were grown in LB broth at 28 or 37 °C to OD_600_ 0.5 or 0.8, and after the addition of 1 mM IPTG, the growth was continued for 4 h more to express a given gene from a T7 promoter in pET-derived recombinant plasmids. Recombinant His-tagged proteins were purified on a Ni^+2^-NTA resin with imidazole (Sigma, Saint-Quentin-Fallavier, France). To assess the level of synthesized RNA polymerase subunits in cells used for the preparation of S30 extracts, protein samples were separated with SDS-PAGE and analyzed by Western blot using for comparison the purified *E. coli* RNA polymerase (Epicentre Technologies, Thane, India). Nitrocellulose membranes with transferred proteins were incubated with 1:200 diluted monoclonal antibodies against α, β or β’ (Neoclone, Madison, WI, USA) under standard conditions. Membranes were washed, incubated with Alexa Fluor 680-labeled goat anti mouse antibody (Eurogentec, Liege, Belgium) for 1.5 h and, after washing, were scanned with Odyssey scanner (Li-cor Biosciences, Bad Hamburg, Germany).

### 2.6. Purification of E. coli RNA Polymerase Subunits from Living Cells

We used the method of obtaining core enzyme RNA polymerase of *E. coli* by purifying inclusion bodies formed in cells carrying four cloned genes together, *rpoA*, *rpoB*, *rpoC* and *rpoZ* [23]. Four subunits, α, β, β’ and ω, which formed inclusion bodies due to overexpression in the same cells, were denatured with 6 M guanidine chloride and renatured by dialysis in Tris-HCl 50 mM, pH 7.9, KCl 200 mM, MgCl2 10 mM, ZnCl_2_ 10 mM, EDTA 1 mM, β-mercaptoethanol 5 mM, glycerol 20%. The subunits were then purified by precipitation with ammonium sulfate followed by chromatography. It was found that the subunit β’ is less present in the resulting purified sample, which can affect the stoichiometry of the subunits required for the formation of the core enzyme. The factor σ^70^, cloned from *rpoD* in His-tag fusion was added to reconstitute the holoenzyme of RNA polymerase. The analysis of the purified enzyme was performed using the “Protein 200 plus assay” on an Agilent 2100 bioanalyzer (Agilent Technologies, Les Ulis, France).

### 2.7. Preparation of S30 Cell-Free Extracts

Bacterial cell-free extracts were prepared by the method described previously [24] with modifications. Cells were grown at 37 °C to OD 0.8, harvested by centrifugation, washed twice thoroughly in ice-cold buffer containing 10 mM Tris-acetate pH 8.2, 14 mM Mg-acetate, 60 mM KCl, 6 mM β-mercaptoethanol and then disrupted by French press (Carver, Belle Plaine, MN, USA) at 9 tones (≈20,000 psi). The disrupted cells were centrifuged at 30,000× *g* at 4 °C for 30 min, and the supernatant was centrifuged again. The clear lysate was added in a ratio of 1:0.3 to the preincubation mixture containing 300 mM Tris-acetate at pH 8.2, 9.2 mM Mg-acetate, 26 mM ATP, 3.2 mM dithiothreitol, 3.2 mM L-amino acids and incubated at 37 °C for 80 min. The mixed extract solution was centrifuged at 6000× *g* at 4 °C for 10 min, dialyzed against a buffer containing 10 mM Tris-acetate pH 8.2, 14 mM Mg-acetate, 60 mM K-acetate, 1 mM dithiothreitol at 4 °C for 45 min with 2 changes of buffer, concentrated up to 4 times by dialysis against the same buffer containing 50% PEG-20,000, followed by additional dialysis without PEG. The obtained S30 cell-free extract was aliquoted and stored at −80 °C.

### 2.8. Cell-Free Protein Synthesis

The batch-mode of cell-free synthetic reaction was carried out under the conditions described by Pratt [25] with some modifications. The standard premix contained 50 mM Tris-acetate pH 8.2, 46.2 mM K-acetate, 0.8 mM dithiotreitol, 33.7 mM NH4-acetate, 12.5 mM Mg-acetate, 125 μg/mL tRNA from E. coli (Sigma), 6 mM mixture of CTP, GTP and TTP, 5.5 mM ATP, 8.7 mM CaCl2, 1.9% PEG-8000, 0.32 mM L-amino acids, 5.4 μg/mL folic acid, 5.4 μg/mL FAD, 10.8 μg/mL NADP, 5.4 μg/mL pyridoxin and 5.4 μg/mL para-aminobenzoic acid. The enzyme pyruvate oxidase was used to generate ATP via acetyl phosphate formation from pyruvate and inorganic phosphate [26] by addition of 32 mM pyruvate in 6.7 mM K-phosphate pH 7.5, 3.3 mM thiamine pyrophosphate, 0.3 mM FAD and 6 U/mL pyruvate oxidase (Sigma). To express the cloned genes from T7 promoter-driven constructs, the T7 RNA polymerase (Promega, Charbonnières-les-Bains, France) was added to the reaction mixture. Circular plasmid DNA or linear PCR-amplified DNA was added to the premix containing all necessary L-amino acids except methionine, replaced by 10 µCi of [α35S]-L-methionine with specific activity 1000 Ci/mmol, 37 TBq/mmol (GE Healthcare, Parçay Mesley, France), and the reaction was initiated by adding *E. coli* cell-free extract followed by incubation at 37 °C for 90 min. The plasmid pET-pargC-8-amaA [27] and pET-pT7-amaA have the same size between the ribosome binding site of promoters and the T7 terminator region, including the 8-nucleotide segment between the Shine–Dalgarno site and the initiation codon for translation. Similar conditions were used for protein synthesis with each cell extract, and the same volume was loaded into lanes for each given protein. Quantification of cell-free synthesized proteins was performed by counting the radioactivity of ^35^S-labeled protein bands with Phosphor Imager 445 SI (Molecular Dynamics, Chatsworth, CA, USA) or by comparing non radiolabeled protein bands with known reference proteins after coloration with EZBlue gel staining reagent (Sigma, Saint-Quentin-Fallavier, France).

### 2.9. Mobility-Shift Assay 

Functional status of cell-free synthesized proteins was tested by mobility-shift assay using promoter-operator regions of *T. maritima* and *E. coli.* A putative promoter-operator region was obtained by PCR using IRDye-labeled primers (Appendix A) as described previously [28]. His-labeled proteins were diluted in binding buffer 10 mM Tris-HCl (pH 7.5), 250 mM KCl, 5 mM MgCl_2_, 2.5 mM CaCl_2_, 2.5% glycerol,0.5 mM DTT and then incubated labeled target DNA in the presence of a 100-fold of unlabeled sonicated herring sperm DNA at 37 °C for 30 min. Samples were loaded on a 2% (*w*/*v*) agarose gel prepared in TAE buffer (40 mM Tris-base (pH 8), 10 mM sodium acetate, 1 mM EDTA) and electrophoresed at room temperature at 12 V cm^−1^ for one hour. DNA fragments were transferred from gels onto a nylon membrane (Qiagen). The concentration of labeled DNAs was measured with a UV/VIS spectrometer (Perkin Elmer, Villebon-sur-Yvette, France) and by comparison of fluorescent DNA bands in an agarose gel.

## 3. Results

### 3.1. Protein Synthesis in S30 Extracts Prepared from Cells Overexpressing RNA Polymerase Subunits

The *G. stearothermophilus* p*argC* promoter possesses conserved −10 and −35 sites separated by a 17-bp spacer and an AT-rich UP element [29], (see Figure 1A). In order to gain higher gene expression from the p*argC* promoter, we decided to increase the pool of the bacterial RNA polymerase in the reaction mixture by using the cell-free extracts in which core enzyme subunits were overexpressed. We constructed recombinant pDuet-based plasmids allowing T7 promoter-mediated coexpression of four genes coding for RNA polymerase α, β, β’ and ω subunits in *E. coli* cells after IPTG induction (see Figure 1B).

Analysis of the supernatant fraction from crude extracts of *E. coli* BL21 Star (DE3) (pETDuet-*rpoA*-*rpoB*/pACYCDuet-*rpoC*-*rpoZ*) showed that the protein bands corresponding to α, β and β’ subunits were more intense than those of the plasmidless strain *E. coli* BL21 Star (DE3). Western blot analysis allowed us to assess the quantity of synthesized RNA polymerase subunits in a soluble fraction of cell extracts prepared from the same lysates (Figure 2). A weak band of the His-tagged α subunit was detected in noninduced cells; however, the amount of α subunit increased nearly 8-fold after IPTG induction. Meantime, the amount of both β and β’ increased about 2- and 2.5-fold, respectively.

To understand the reason for this difference, we performed a similar analysis of the insoluble fraction of the same extracts. It was observed that the majority of β and β’ subunits and, to a lesser extent, α subunits, precipitated in the pellet. Protein bands from cultures induced at OD_600_ 0.5 and 0.8 have almost similar intensity, suggesting that the prolonged exponential growth before the IPTG-induction does not improve the yield of a soluble fraction of bacterial RNA polymerase subunits. Notably, the formation of an insoluble protein, probably in the form of β’ subunit inclusion bodies, was also observed in living cells when four subunits of RNA polymerase were expressed together in the same cells. The addition of separately purified σ factor likely improved the solubility and slightly increased yield of the holoenzyme RNA polymerase, comparable to a commercial enzyme (Figure 3). To increase the yield of RNA polymerase subunits, we recovered the various subunits by renaturation from inclusion bodies denatured in 6 M guanidine chloride (Appendix A). However, the purified complex contained mainly α_2_β and a low amount of β’.

Next, we compared the p*argC*-mediated aminoacylase expression in cell-free extracts prepared from *E. coli* BL21 Star (DE3) (pETDuet-*rpoA*-*rpoB*/pACYCDuet-*rpoC*-*rpoZ*) cells using pETp*argC*-*amaA* circular plasmid DNA (Figure 1C) as a template at two concentrations. In two independent experiments, the amount of aminoacylase synthesized on circular plasmid DNA from the T7 phage promoter was quite close for DNA concentrations of 8 and 55 ng/μL (Figure 4). Meanwhile, the protein yield on this construct was almost double that of the pargC-amaA plasmid at 55 ng/μL DNA after IPTG-induction of cells at OD 0.8 (Figure 4B,D). We also compared both promoters in cell-free aminoacylase synthesis at OD 0.8 after subtracting the protein yield for *E. coli* BL21 Star (DE3) were incubated with IPTG. The pT7 promoter provided approximatively 5.3 and 3.8 times more protein at DNA concentrations of 8 and 55 ng/μL, respectively, than the pargC-8 promoter at the same conditions.

We observed a relatively higher level of aminoacylase in the extracts from noninduced cells carrying the plasmids encoding the RNA polymerase subunits, compared to extracts from cells lacking these plasmids. This appears to reflect a leaky nature of the T7-based expression system used as proven by a weak band of the His-tagged RpoA in extracts from noninduced cells (lane 2 in Figure 2).

**Figure 3 biotech-10-00024-f003:**
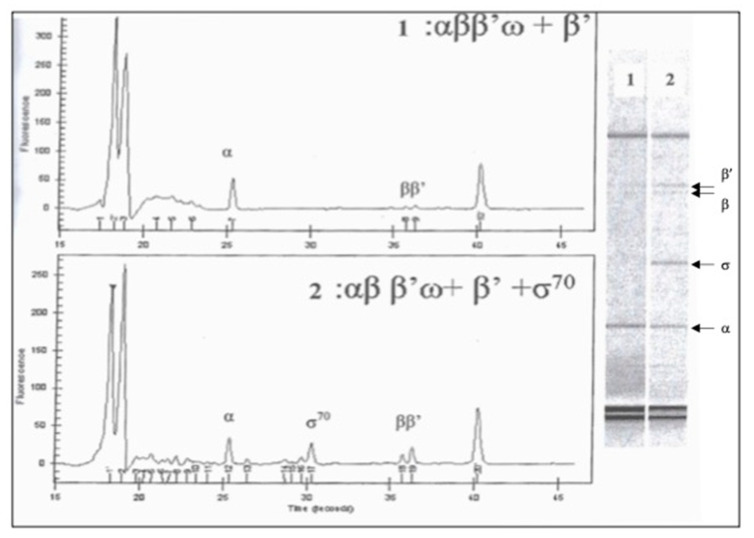
Analysis of *E. coli* RNA polymerase overexpressed in living cells. Detection of RNA polymerase subunits by capillary microelectrophoresis without σ factor (1) and with added σ70 factor (2). Factor σ70 was purified separately and added to the purified proteins obtained in the soluble fraction.

Considering that the bacterial and phage promoters have an 8-nucleotide spacer between the ribosome binding site and the initiation codon for translation, as well as the same region of the T7 terminator, it can be concluded that the pargC promoter itself provides a relatively high yield of a given protein produced in a CFPS system.

### 3.2. Extension of a 3′ Extremity of mRNA Templates and T7 Transcription Terminator Increase CFPS

The presence of stem-loop structures in a 3′ untranslated region (3′ UTR), recognized as transcription termination signals by RNA polymerase, protects nascent mRNAs from 3′-exonuclease degradation [30,31]. Therefore, to enhance the P*argC*-mediated gene expression from linear DNA, we decided to enlarge the target gene mRNA template transcribed from PCR products by conserving genomic sequences located downstream of corresponding coding regions.

**Figure 4 biotech-10-00024-f004:**
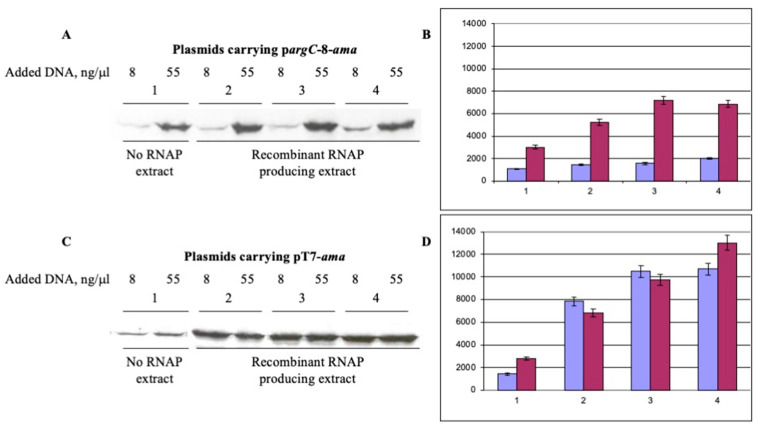
Comparison of aminoacylase synthesis under the control of p*argC* and pT7 promoters in cell-free extracts without and with induction of *E. coli* RNA polymerase α, β, β’ and ω subunits. DNA of plasmids pET-p*argC*-8-*amaA* (**A**,**B**) and pET-pT7-*amaA* (**C**,**D**) was added to the reaction mixture at concentrations of 8 ng/µL (color violet) and 55 ng/µL (color mauve). Lane 1 plasmidless *E. coli* BL21 Star (DE3) incubated with IPTG; lanes 2, noninduced *E. coli* BL21 Star (DE3) (pETDuet-*rpoA-rpoB*/pACYCDuet-*rpoC-rpoZ*); lane 3, *E. coli* BL21 Star (DE3) (pETDuet-*rpoA-rpoB*/ pACYCDuet-*rpoC-rpoZ*) induced with IPTG at OD_600_ 0.5; lane 4, *E. coli* BL21 Star (DE3) (pETDuet-*rpoA-rpoB*/ pACYCDuet-*rpoC-rpoZ*) induced with IPTG at OD_600_ 0.8. The incorporation of ^35^S methionine into the synthesized aminoacylase was assessed using a PhosphorImager. Histograms represent data from two experiments.

Linear DNA fragments, corresponding to *G. stearothermophilus argC* and *amaA* genes and *T. maritima gntR0439* gene, were amplified by PCR by conserving 3, 100 or 200 bp sequences located downstream of the termination codon of each gene. A similar, rather high protein yield was found for GntR0439 from all three linear DNAs (Figure 5A) suggesting that the transcribed *gntR0439* mRNA covering the only coding region is a good substrate for bacterial ribosomes to provide high protein productivity. By contrast, two other genes, *argC* and especially *amaA*, required longer DNA templates to be translated into the corresponding polypeptides (see Figure 5A). If a 3 bp extension was enough to obtain an abundant band of a 38 kDa *N*-acetylglutamate-5-semialdehyde dehydrogenase encoded by *argC*, a longer 100 bp genomic sequence was necessary to achieve a high synthesis of a 42 kDa aminoacylase encoded by *amaA*. Further elongation of the respective regions located downstream had less effect on the expression of both genes. These data suggested that appending a 100–200 bp sequence downstream of target genes by PCR amplification from bacterial genomes is beneficial and could potentially be used to protect corresponding mRNAs from a ribonuclease attack.

We studied the effect of a well-characterized T7 phage stem-loop structure on the *amaA* reporter gene expression from p*argC*. Two linear DNA fragments were prepared, composed of the p*argC* region linked to the *G. stearothermophilus amaA* gene with its own 200 bp 3′ UTR or connected to the same gene with a 48 bp transcription terminator region of the pET3 vector used for the construction of pETp*arg*-*amaA* (see Figure 1C). PCR amplified higher aminoacylase expression from linear DNA containing the phage stem-loop than from the analogous construct containing a 200 bp 3′ UTR at DNA concentration 55 ng/μL. Moreover, the p*argC* promoter module combined with the phage transcription terminator stimulated the aminoacylase productivity from a linear DNA to a level close to that of a circular plasmid DNA.

### 3.3. Protein Synthesis Is Higher in RNase-I-Deficient Cell-Free Extracts

The *E. coli* BL21 (DE3) host, naturally devoid of OmpT endoprotease and Lon protease [32], is better suited for CFPS [33]. The mutation *rnaE131* in *E. coli* BL21 Star (DE3) leading to the formation of a C-terminal truncated RNase E, provides higher expression in cells of cloned genes transcribed from strong T7 phage promoters [34]. Considering that the activity of endoribonuclease RNase I could determine the stability of messages, it was tempting to assess the contribution of this enzyme to the protein production in cell-free extracts of an RNase I-deficient mutant (*rna*::Tn10).

Mutant strains deficient in RNase E (*rnaE131*), RNase I (*rna*::Tn10) and RNase E/RNase I activities were first compared in their endoribonuclease activity with parent strains using commercial substrate from Ambion. The crude lysates of the mutant *rna*::Tn10 exhibited more than two-fold lower activity (Table 1). Meantime, no difference was detected in crude extracts of wild-type *rnaE*^+^ and mutant *rnaE131* cells, suggesting that RNase E does not hydrolyze the used substrate. The endoribonuclease activity remained almost at the same ratio in crude and cell-free extracts prepared from lysates of wild type *rna*^+^ and mutant *rna*^−^ cells.

Next, we compared cell-free synthesis of prokaryotic and eukaryotic proteins using the extracts prepared from RNase I-deficient and -proficient strains. The reaction was initiated by adding T7 RNA polymerase to the mixture containing a T7 promoter expression plasmid carrying bacterial genes *amaA*, *groES* and *rraA* or human genes *nat1* and *nat2*. Two independent preparations of S30 extracts were tested and gave similar positive results regarding the effect of RNase I deficiency. Indeed, autoradiography of gels clearly showed that protein bands were stronger with the extracts of *E. coli* BL21 (DE3) *rna*::Tn10 and *E. coli* BL21 Star (DE3) *rna*::Tn10 rnaE131 mutant strains than with the extracts of the respective rna+ hosts (Figure 5B). As judged from the measurement of the incorporated [α35S]-L-methionine, the extracts of RNase I-deficient cells provided 2–4-fold higher yields of the tested proteins, and this difference was the most pronounced for the bacterial GroES chaperone.

The *amaA* gene expression was also evaluated with p*argC*-*amaA* DNA in S30 extracts of RNase I-proficient and -deficient strains. The p*argC* promoter provided almost two-fold lower yield of aminoacylase than the pT7 promoter, as estimated by relative incorporation of a radioactive methionine into the protein. However, an increase in protein synthesis was again detected in the extracts of both *rna*::Tn10 and *rnaE131 rna*::Tn10 strains (see Figure 5B), suggesting that a positive effect of the RNase I deficiency is not dependent on the rate of mRNA synthesis in cell-free extracts. Surprisingly, no difference was detected in the yield of bacterial or human proteins synthesized in S30 extracts of *rna*^+^ and *rnaE131* bacteria, suggesting that the corresponding mRNAs are not susceptible to cleavage by RNase E.

**Figure 5 biotech-10-00024-f005:**
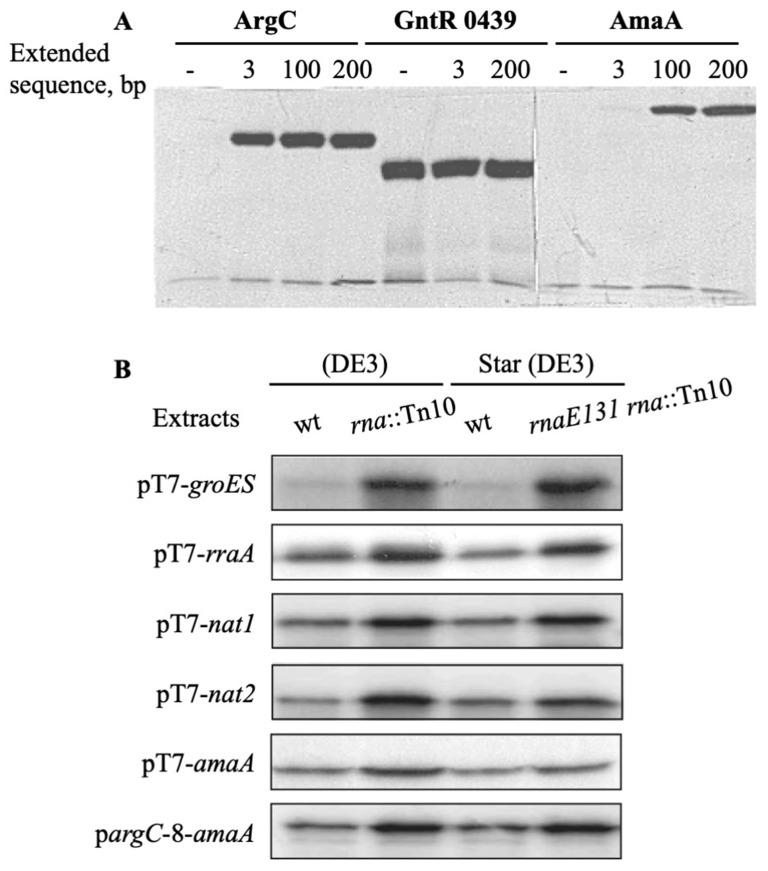
Effect of 3′-UTR extension on the p*argC*-promoted gene expression and protein production in wild-type and mutant RNase I (*rna*::Tn10) and RNase E (*rnaE131*) strains of *E. coli*. (**A**)—Lanes 1, 2, 3 and 4: *G. stearothermophilus argC* DNA, respectively, without and with an extended 3, 100 or 200 bp genomic sequence; lanes 5, 6 and 7: *T. maritima gntR 0349* DNA, respectively, without and with an extended 3 or 200 bp genomic sequence; lanes 8, 9, 10 and 11: *G. stearothermophilus amaA* DNA, respectively without and with an extended 3, 100 or 200 bp genomic sequence. (**B**)—The *T. neapolitana groES*, *E. coli rraA*, human *nat1* and *nat2* genes were expressed from pT7 promoter in pET-pT7-*x* plasmids (*x*—any gene); and *G. stearothermophilus amaA* gene was expressed on pETpT7*-amaA* and pETp*argC-amaA* plasmids. Lanes 1–4, with S30 extracts of *E. coli* BL21 (DE3), *E. coli* BL21 (DE3) *rna*::Tn10, *E. coli* BL21 Star (DE3) *rnaE13* and *E. coli* BL21 Star (DE3) *rna*::Tn10 *rnaE131* strains, respectively. The sample volume used for analysis with pargC-8-amaA was twice that of plasmid pT7-8-amaA. Similar results were obtained in two assays.

**Table 1 biotech-10-00024-t001:** Endoribonuclease activity in engineered *E. coli* strains.

RNase Status	Relative Endoribonuclease Activity (%)	
	Crude Extracts	S30 Extracts
BL21 (DE3)	100	100
BL21 (DE3) *rna*::Tn10	42	51
BL21 Star (DE3) (it is *rna*E131 strain)	113	92
BL21 Star (DE3) *rna*E131 *rna*::Tn10	38	48

### 3.4. DNA–Protein Interaction in Cell-Free System

The production of a functionally active protein in a cell-free system is an important criterion for the quality of the developed method. In this study, we evaluated the interaction of synthesized regulatory proteins in binding ability to a target DNA. Similar DNA-binding sequences in evolutionary distant bacteria are indicative of the resemblance between regulatory systems. Therefore, in the absence of adequate information on protein–DNA interactions, we tested our cell-free system using heterologous combination of interaction partners.

Three proteins belonging to Gnt, Xyl and Lac families of *T. maritima* putative regulators were cell-free synthesized and probed to IRDye-labeled operator sequences of *E. coli*, namely *gntKo* [35], *xylFo* [36] and *lacIo* [37]. Moreover, the cell-free synthesized *T. neapolitana* ArgR repressor was also probed to *argRo* DNA carrying a regulatory region of the same bacterium.

Mobility-shift analysis revealed that target DNAs bind to two proteins, *T. neapolitana* ArgR and *T. maritima* Gnt 0439 evidenced by the formation of clear retarding bands in the agarose gel (Figure 6). The putative *lacIo* operator binding is likely to occur also with Lac 1856 as shown by a decrease in the protein band intensity in the presence of the target DNA. By contrast, no DNA binding was detected for the Xyl 0808 protein, suggesting that heterologous DNA from *E. coli* is not an interaction partner with the hyperthermophile protein. These results indicated that at least some of the transcriptional regulation proteins synthesized in the cell-free system are correctly folded to bind to target DNA probes.

## 4. Discussion

The development of alternative expression modules remains an important challenge in expanding CFPS for biomedical applications and high-throughput proteomic studies. A retrospective evaluation of CFPS approaches highlights the importance of improving the transcriptional machinery to produce various functionally active proteins [38]. The efficiency of transcription determines the processing of mRNA by the rate-limiting step for CFPS.

In our study, several approaches were tested to improve the transcription process and increase cell-free production of different proteins. We demonstrated that the *G. stearothermophilus* p*argC* promoter (see Figure 1) provides rather high gene expression in cell extracts prepared from *E. coli* BL21 (DE3). The strong promoter p*argC* enhances up to 1.8-fold the target protein yield in cell-free extracts prepared from bacteria, in which the cloned genes coding for α, β, β’ and ω subunits of *E. coli* RNA polymerase are simultaneously coexpressed from two compatible plasmids. The limited capacity of the coexpressed RNA polymerase subunits to support the high transcription of the target *amaA* gene from the strong p*argC* promoter could be related to the insolubility of β and β’ subunits and/or to an insufficient quantity of a major σ^70^ factor critical for the assembly of a greater amount of a functionally active holoenzyme.

We have shown that the extension of the *amaA* gene, by 100–200 bp segment located downstream, can substantially improve the p*argC*-mediated expression from linear DNAs, suggesting the participation of a stem-loop structure within the corresponding 3′ UTR. Indeed, such a structure of the T7 transcription terminator remarkably enhances the *amaA* gene expression from a linear template, leading to a protein yield close to that observed with circular plasmid DNA. Similar results have been obtained for pT7-mediated gene expression [39,40]. Therefore, linear DNAs carrying potential stem-loop forming sequences at the end of the coding region or within 3′ UTR can contribute to enhancing CFPS independently of the nature of the promoter used.

The difference in the endo- and exo-ribonuclease activities of individual enzymes determines the ability of the transcriptional degradasome to provide different levels of functionally active mRNAs in cells. The diversity and complexity of mechanisms controlling the quantity and quality of mRNAs, which vary in their length, primary sequence, and secondary structure, make the development of a universal cell-free system that ensures the high production of numerous proteins in parallel reactions a difficult task. RNase E appears to play a major role in mRNA decay in the bacterial cytoplasm. A 2 nt deletion of the *rnaE* gene (*smbB131* mutation renamed *rnaE131*) causes a premature stop of translation at the 584th codon of RNase E, which stabilizes mRNAs susceptible to hydrolysis by RNase E [7]. Consequently, *E. coli* BL21 Star (DE3) cells harboring the *rnaE131* mutation produce more proteins from RNase E sensitive messages [34]. The preservation of mRNA degradation has been also shown in RNase E-deficient cell extracts of *E. coli* [41]. However, our data indicate that the synthesis of bacterial AmaA, GroES and RraA or human NAT1 and NAT2 proteins is not increased in the *rnaE131* cell-free extracts of *E. coli* BL21 Star (DE3). In this context, it is worth mentioning that RNA decay and processing appear to occur at specific cell membrane sites that are in contact with a multiprotein degradosome complex via RNase E [42]. During the preparation of S30 extracts, the cell membrane is discarded; therefore, the degradosome complex might be dissociated and consequently, RNase E would not be able to fully accomplish degradosome-related functions. Despite remarkable progress in improving cell-free transcription for protein synthesis, investigations in this direction remain quite current [43].

To assess the action of RNase I endoribonuclease on the stability of mRNAs in a cell-free system, we constructed *rna*::Tn10-deficient mutants of the *E. coli* BL21 strain. RNase I is processed from a precursor protein by cleavage of a leader peptide followed by exporting a 27 kDa enzyme into the periplasm of Gram-negative bacteria [10]. Our data suggest that the liberated enzyme is retained in S30 extracts; therefore, both prokaryotic and eukaryotic genes are better expressed in the extracts prepared from RNase-I-deficient *rna*^−^ mutants rather than RNase-I-proficient *rna*^+^ strains. Obviously, the application of new extracts from *E. coli* BL21 *rna*^−^ mutant cells, which lack the periplasmic EDTA-tolerant RNase I, is a radical way to prevent a rapid decay of messages and thus to improve the performance of cell-free protein production promoted by strong bacterial or phage transcription signals.

We have previously shown that cell-free synthesized proteins allow simultaneous and rapid assessment of binding to many promoters, thereby experimentally confirming the accuracy of in silico prediction of strong promoters in the genome of the hyperthermophilic bacterium *T. maritima* [44]. This study shows that diversifying and integrating bacterial and/or phage transcription signals into cell-free extracts can significantly improve the efficiency of CFPS and increase the yield/production of both prokaryotic and eukaryotic proteins. Moreover, CFPS can overcome the major obstacles faced by hard-to-express proteins in bacterial cells, such as transmembrane proteins, toxic proteins, and aggregation-prone proteins or proteins from bacteria maintained in nongrowing laboratory conditions [45,46].

## 5. Conclusions

Cell-free protein production and its use to study molecular interactions is an alternative approach to cell-based methods commonly employed in different fields of biology. Cell-free extracts from prokaryotic cells have been developed to optimize the conditions for specific protein expression enabling target binding and identification efforts, as well as uncovering novel enzymatic activities. Considering the difficulties of protein production in cells, cell-free systems are being increasingly used to generate proteome scale tools for biomedical research. The advantageous characteristics of cell-free systems described in this and other studies open attractive opportunities for high-throughput and multiplexed screening methods adapted to CFPS, such as protein, antibody and small molecule microarrays, to identify and study new therapeutic agents for various human diseases.

## Figures and Tables

**Figure 1 biotech-10-00024-f001:**
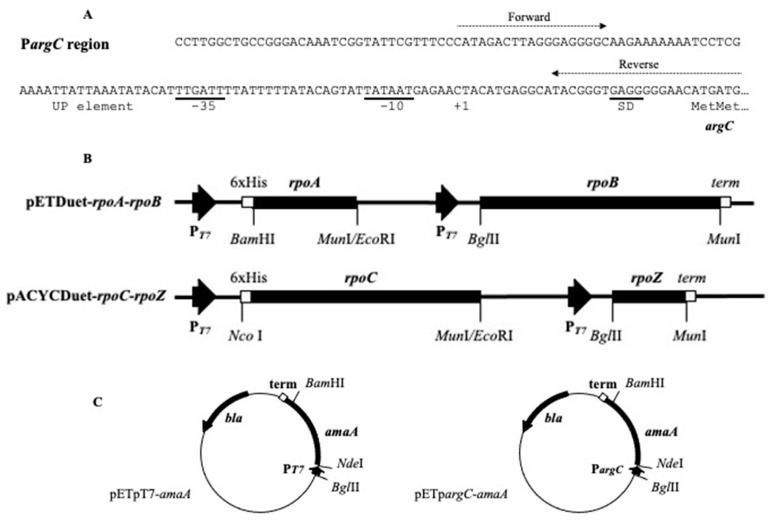
The *G. stearothermophilus* p*argC* region (**A**), the maps of recombinant plasmids carrying the *E. coli rpo* genes for core RNA polymerase (**B**) and the *G. stearothermophilus amaA* reporter gene (**C**). The sequence of the used p*argC* region is shown; dotted arrows show positions of forward and reverse primers used for PCR. In a couple of pET-rpoA-rpoB and pACYC-rpoC-rpoZ plasmids, the *rpoA* and *rpoC* gene code for N-terminal His-tag. Abbreviations: UP for UP-element, SD for the site Shine-Dalgarno, term for T7 transcription terminator.

**Figure 2 biotech-10-00024-f002:**
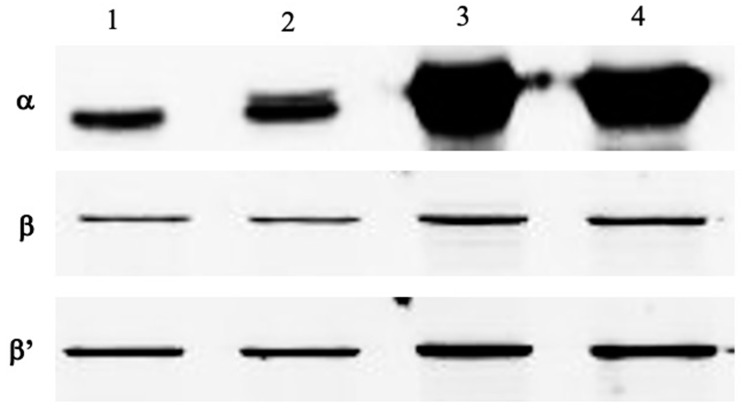
Western blot of RNA polymerase subunits α, β and β’ in the supernatant of S30 extracts of cells without and with coexpression of genes *rpoA*, *rpoB*, *rpoC* and *rpoZ*. Detection of subunits was performed with the corresponding mAb. Lane 1, plasmidless *E. coli* BL21 Star (DE3) incubated with 1 mM IPTG; lane 2, *E. coli* BL21 Star (DE3) (pETDuet-rpoA-rpoB/pACYCDuet-rpoC-rpoZ) incubated without IPTG; lanes 3 and 4, *E. coli* BL21 Star (DE3) (pETDuet-rpoA-rpoB/pACYCDuet-rpoC-rpoZ) induced with IPTG at OD600 0.5 and 0.8 of cell growth, respectively.

**Figure 6 biotech-10-00024-f006:**
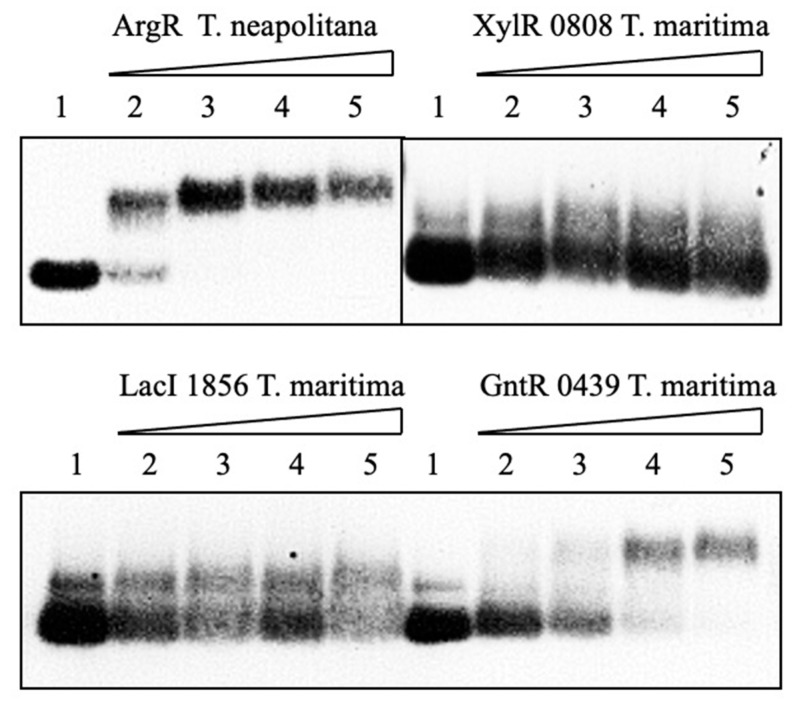
Gel-shift analysis of *T. neapolitana* and *E. coli* DNA operator regions binding to cell-free synthesized *T. neapolitana* ArgR and *T. maritima* putative regulatory proteins. Lane 1—without protein; lanes 2–5 contain 12, 24, 48, and 96 nM operator DNA. His-tagged purified proteins were probed to labeled operator DNA at 37 °C for 30 min and subjected to electrophoresis on a 2% agarose gel. ArgR binding was carried out in the presence of L-arginine in both binding and running buffers.

## Data Availability

The data are available upon request from the authors.

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
