# Peer review of "Cell-Free Protein Synthesis by Diversifying Bacterial Transcription Machinery"

_biotech, 2021, doi:10.3390/biotech10040024_

Round 1

Reviewer 1 Report

The manuscript by Lebon et al. reports on the optimization of cell-free protein synthesis (CFPS) using bacterial transcription machineries. CFPS has certain benefits over in vivo protein synthesis methods commonly utilized in lab settings and therefore, optimization/betterment of CFPS systems can have a broad implication in standard molecular biology/protein biochemistry and biotechnology. While the study focuses on a crucial question, results presented in the study lacks certain experimental rigor that is expected to convince its readers. A list of concerns (from this reviewer’s perspective) are listed below:

  1. Proper statistical quantifications should be performed to compare effects of different plasmid modifications. For example, Figs. 4 and 5 show qualitative differences between different plasmids/constructs. Such qualitative differences should be quantified to check whether the observed differences are statistically significant.
  2. Related to point 1, the authors should perform(/specify) independent biological replicates for each plasmids/conditions so that one can clearly distinguish effects of promoter regions, DNA concentrations on the CFPS efficiency.
  3. While the authors claim that pargC promoter is better in CFPS compared to pT7, it is not clear from the presented data that this is indeed the case. Quite the contrary, based on Fig. 4B and C, an argument can be made that pT7 works better.
  4. In section 4, the authors should discuss the possibility that their CFPS could have protein folding issues. Typically, many of the DNA-binding domains are capable of non-specific binding. Lowering salt conc. might help in detecting such interactions. The authors also did not provide experimental details for the DNA binding assay. This protocol should be provided.
  5. Minor: Instead of labeling gel lanes with numbers, maybe the authors can just provide complete labels for individual gel lanes. That would be easier for the readers (esp. since the journal doesn’t have any page limit and figures in the manuscript have a lot of void space).

Author Response

We thank the reviewer for her/his comments, which helped improve our manuscript. We addressed all the concerns. Note, our modifications, including new references, are in red color. We also introduced new references. Please see answers below in italic.

The removal of author 1 was made at his request in accordance with the legislation and with the consent of the journal.

Reviewer 2 Report

Here we have an interesting attempt to improve one of the cell-free protein synthesis systems. That could be useful for some biotechnological processes, and, according to the words of the authors, for high-throughput screening assays. Many experiments have been performed, and some promising results have been obtained. Nevertheless, the authors honestly said to us that, for example, the synthesis of bacterial AmaA, GroES and RraA or human NAT1 and NAT2 proteins was not increased in the rnaE131 cell-free extracts of E. coli BL21 Star (DE3). Additionally, the authors failed to get some proteins folded properly, and this indicates us that artificial obtaining of any new protein still remains a mysterious process, a work of art.

According to the manuscript, I have some questions and remarks which, to my opinion, should be answered and/or corrected before publication.

  • Generally, I want to ask the ultimate question (of life, the universe, everythingJ). How justified is it to make such a complex system? Is it really more profitable than selecting conditions for the intracellular synthesis of proteins of interest? I agree that some proteins is pretty hard to get, and, from time to time, even impossible. But what about another ones? If we are talking about benefits of avoiding toxic effects to the host system during synthesis of heterologous proteins, it would be honest to mention opportunities which the pBAD vectors can provide us. And what about the effect of the production of several protein complexes at once in microorganisms? Does this not affect their vital functions and the volume of the output of the cell extract? Please discuss it additionally in the Discussion chapter.
  • Please let us see a table with primers which were used to construct all the vectors listed in lines 69-85. It can be placed to the Supplementary materials, for example.
  • Lines 109-111: how exactly the transfer was performed? Please describe the process more strictly.

Author Response

We appreciate the reviewer’s feedback on our manuscript. We addressed the reviewer’s comments and added the modifications in the corresponding sections of the paper. Note, our modifications, including new references, are in red color. Please see answers below in italic.

The removal of author 1 was made at his request in accordance with the legislation and with the consent of the journal.

Reviewer 3 Report

The paper presented by Lebon and colleagues describes a lor of effort aimed to improve cell free protein synthesis.  The paper is well written, but some control and methods are missing. In particular, the introduction section should be implemented including more recent papers/reviews (such as PMID: 32775710) describing the more recent strategies developed to improve E. coli CFPS.  

In methods section several details are missing:

- please specify the source of Nat1 and Nat2 cDNA: human cells, commercial library?

- please add a table containing the sequence of all used primers.

- No information about sequence verification are present; agarose gel figure of the PCR amplification can be added.

- A purification protocol is absent: please add it.

- At line 139 subscript MgCl2

- A mobility-shift analysis protocol must be added.

In the results section:

- A figure of the coomassie stained SDS-PAGE of Fig. 2 must be added. To better show the solubility degree of the RNA polymerase subunits, the gel could contain both supernatant and pellet fractions.

- Please explain the sentence at lines 211-212 "indicating that the prolonged exponential growth before the IPTG-induction does not affect the yield of a soluble fraction of bacterial RNA polymerase subunits". Why do you expect an increase in solubility delaying the induction?

- Please clarify the phrase at lines 216-217: when the σ-factor is added? If the answer is during cell free synthesis, a picture of the soluble and insoluble fraction can be added to compare the results showed in the new figure 2.

- In figure 3 please indicate the protein immediately under σ70.

- In figure 4 a full-size image with molecular weight protein marker should be added.

- In figure 5A, a molecular weight protein marker should be added.

- In figure 5B, please indicate the total µg of protein loaded in each lane.

- In figure 6 a better description of the lanes should be added in the legend such as a molecular weight marker.

Author Response

We thank the reviewer for the thoughtful comments. We have made every effort to address these. Please find below our answers to your comments in italic. We have expanded the introduction section and added new references. Note, our modifications and new references are in red color.

The removal of author 1 was made at his request in accordance with the legislation and with the consent of the journal.

Round 2

Reviewer 1 Report

The revised manuscript addresses the concerns 3 to 5 of this reviewer. However, it is not clear how concerns 1 and 2 has been addressed. The author response letter simply mentions that “We have added the necessary information in Figures 4 and 5 and legends”.

This reviewer had concerns with regards to the number of replicates for each plasmids presented in Figs. 3 and 4. Was this done? The error bars in Fig. 4B and 4D – are they errors from multiple experimental replicates? If yes, the number of replications should be provided as n=?.

Another concern (Concern 1 in the prev. submission) was the statistical significance of comparison between different promoters. This would be addressed if experimental replicates are performed and different protein production levels are compared using statistical methods to see whether the observed differences are indeed statistically significant or they are within the variations of measurement methods employed.

Author Response

The revised manuscript addresses the concerns 3 to 5 of this reviewer. However, it is not clear how concerns 1 and 2 has been addressed. The author response letter simply mentions that “We have added the necessary information in Figures 4 and 5 and legends”.

Response: Thank you very much for your critical comments.

Our corrections are in red color. I think we have tried to respond to your suggestions.

This reviewer had concerns with regards to the number of replicates for each plasmids presented in Figs. 3 and 4. Was this done? The error bars in Fig. 4B and 4D – are they errors from multiple experimental replicates? If yes, the number of replications should be provided as n=?.

Response: In the previous version, it was mentioned that the data in Fig. 4 were obtained from two experiments. In this version, we followed your suggestion and included this information in the text (line 261) and legend in Fig. 4(line 302), as well as the legend in Fig. 5 (line 378). I think you mean this figure, and no Fig. 3.

Another concern (Concern 1 in the prev. submission) was the statistical significance of comparison between different promoters. This would be addressed if experimental replicates are performed and different protein production levels are compared using statistical methods to see whether the observed differences are indeed statistically significant or they are within the variations of measurement methods employed.

Response: We agree with you. Unfortunately, due to the pandemic and other reasons, it was impossible to conduct three experiments. Therefore, we propose conditional data on the strength of two promoters in a cell-free system, obtained in two analyzes (lines 265-269).

Reviewer 3 Report

I appreciate the revised version of the paper

Author Response

Dear Reviewer 3,

Thank you very much for reviewing our manuscript.